# Malignant Catarrhal Fever in Sardinia (Italy): A Case Report

**DOI:** 10.3390/vetsci9080442

**Published:** 2022-08-19

**Authors:** Elisabetta Coradduzza, Rosario Scivoli, Davide Pintus, Angela Maria Rocchigiani, Maria Giovanna Cancedda, Daria Sanna, Simona Macciocu, Fabio Scarpa, Roberto Bechere, Giantonella Puggioni, Ciriaco Ligios

**Affiliations:** 1Istituto Zooprofilattico Sperimentale della Sardegna, 07100 Sassari, Italy; 2Dipartimento di Scienze Biomediche, Università di Sassari, 07100 Sassari, Italy

**Keywords:** malignant catarrhal fever, ovine herpesvirus type 2 (OvHV-2), calf

## Abstract

**Simple Summary:**

Malignant Catarrhal Fever is a globally distributed disease that is fatal to susceptible species such as cattle. Sheep represent the reservoir species, and the Mediterranean island of Sardinia, which hosts a large number of these animals, is one geographic area where virus can easily spread. The aim of our study was to investigate a case of Malignant Catarrhal Fever in a calf, also studying the prevalence of the virus responsible, Ovine Herpesvirus type 2, among sheep in Sardinia to further investigate the epidemiological aspects. The analyses performed were consistent among each other; indeed, the histological analysis revealed patterns of lesions, which are commonly reported in literature, in many tissue samples of the calf object of the study. We also found a considerable number of copies of viral genomes in all examined organs of the animal. Phylogenetic analyses suggested the possible occurrence of a unique genetic cluster that is widely distributed across the whole Italian territory. In conclusion, the present study provides a comprehensive overview on the Malignant Catarrhal Fever in an area where, despite the high prevalence of the Ovine Herpesvirus type 2 found among sheep, the sporadic occurrence of clinical disease in bovine should be still deeply investigated.

**Abstract:**

Using a multidisciplinary approach, this report describes a clinical case of malignant catarrhal fever (MCF) occurring in a calf, which shared the pasture with sheep on a farm located in the island of Sardinia (Italy). We confirmed the conventional clinico-histopathological features of MCF, as well was the presence of Ovine herpesvirus type 2 (OvHV-2) DNA in several tissues, employing histological and virological investigations. The phylogenetic analysis revealed that this Sardinian OvHV-2 strain is genetically similar to all the other Italian strains. By Real Time PCR examinations of blood samples collected across Sardinia’s sheep population, which is considered the most important reservoir species, we discovered an OvHV-2 prevalence ranging from 20 to 30 percent. Despite the high prevalence of OvHV-2 in the Sardinian sheep population, clinical disease in bovine remains sporadic; further investigations are needed to understand the risk factors that regulate this epidemiological aspect.

## 1. Introduction

Sheep-associated malignant catarrhal fever (MCF) is caused by ovine herpesvirus type 2 (OvHV-2), a virus belonging to the Macavirus genus of the Herpesviridae family [1]. OvHV-2 naturally occurs as a subclinical, lymphotropic infection in reservoir hosts, while it causes a clinical disease in susceptible hosts such as cattle, deer, bison, swine, giraffe, and antelope [2]. MCF has been reported for the first time in Europe, but it is now distributed worldwide [2] and it is most commonly transmitted through contact with reservoir species that shed virus particles in nasal and ocular discharges [3,4]. OvHV-2 is thought to enter the body primarily through inhalation; however, ingestion may also be possible [1]. In addition, the virus can be transmitted vertically during the fetal period through the placenta, although that it is considered to be uncommon [2]. Clinical signs include fever, lymphadenopathy, anorexia, sialorrhea, keratoconjunctivitis, corneal opacity, depression, diarrhea, and neurological signs [3,5]. Normally, for wild ruminants, death occurs within a few days, whereas cattle can survive for several weeks and recover [2]. The disease is notable for affecting only one or a few heads in a herd [2]. Only a few MCF reports have been reported in Italy to date [6,7,8,9,10,11,12] and, to the best of our knowledge, MCF has never been reported in Sardinia. In this article, we describe the clinical, histopathological, virological, and phylogenetical aspects of MCF in a calf and we assess the prevalence of the virus in the Sardinian sheep population to gain a new insight into the disease’s epidemiology.

## 2. Materials and Methods

### 2.1. Case and Sampling

Our report details a case of MCF occurring in a six-month-old Brown calf that happened on a farm in Northern Sardinia (Italy), where sheep also shared the pasture. Following the appearance of clinical symptoms, serum, EDTA samples, and nasal swab were collected for diagnostic investigations. Due to the severe clinical conditions, the calf was euthanized and submitted to necroscopy. The brain, thymus, lungs, small intestine, kidney, and liver were collected for histopathological, virological and bacteriological analysis. We were unable to detect the presence of OvHV-2 in the sheep sharing the pasture with the infected calf and for this reason, we aimed to outline the epidemiological situation of OvHV2 in Sardinia, where ovine farming is an important zootechnic activity with a total population of 3,000,000 sheep. The survey involved 1160 sheep reared in 24 different farms, spread across the island: 6 for the province of Sassari, 2 for Oristano, 7 for Nuoro, and 9 for Cagliari. EDTA blood samples were collected from healthy sheep with standard procedures.

### 2.2. Histopathology

For histological examinations, tissue samples were fixed in 10% neutral formalin. By using a routine protocol, paraffin embedded sections were obtained from each organ, cut into 4-µm-thick slices, and then stained with ST Infinity Hematoxylin & Eosin Staining System (Leica). The stained sections were examined using a light microscope with a magnification range of 50× to 400×. Furthermore, to define the local immune response related to MCF, CD immunophenotyping of inflammatory cells were carried out on selected kidney, liver, and brain tissue sections using primary antibodies CD3+ (DAKO AGILENT), CD79 (DAKO AGILENT) and PAX 5 (DAKO AGILENT), CD68 (BIORAD) and CD163 (BIORAD) specific for T lymphocytes, B lymphocytes, and macrophages, respectively. Sections were heat-induced in citrate buffer (pH 6.0) (Thermo Scientific, Waltham, MA, USA) to expose antigen epitopes, then slides were immersed in a solution of 0.3% H_2_O_2_ for 30 min at room temperature to quench endogenous peroxidase activity. The primary antibodies were incubated overnight at 4 °C. Immunohistochemistry (IHC) was performed using Dako EnVision kit (Dako Agilent, Santa Clara, CA, USA) and counterstaining slides with Mayer’s haematoxylin. Each IHC run included adequate positive control tissues.

### 2.3. Bacteriological Analysis

Bacteriological examinations were performed by culturing on blood agar for *Pasteurella* spp. and *Streptococcus* spp. The ocular swab taken prior to the animal’s death was subjected to a culture test.

### 2.4. Virological Analysis

Direct immunofluorescence techniques (IFA) using specific polyclonal sera (Kit IFA EUROKIT S.r.l.) were used to screen lung tissue for discarding Bovine Viral Diarrhea Virus (BVDV), Bovine Herpes Virus 1 (BHV1), Bovine Herpes Virus 4 (BHV4), Bovine Respiratory Sincytyal Virus (BRSV), and Bovine Parainfluenza 3 (PI3) virus infections. The fluorescence was evaluated using an optical fluorescence microscope (Leica, Mod. DM RBE). Nucleic acid purification from thalamus, lung, liver, kidney, and intestine was performed using MagMax Core Nucleic Acid Purification Kit (Applied Biosystems-ThermoFisher Scientific-Waltham, MA, USA) in an automated sample preparation workstation MagMAX Express 96 (Applied Biosystems), according to the manufacturer’s instructions. Real Time PCR was used to detect the OvHV-2 genome in calf tissues and during the sheep screening whole blood samples, as described by D. Hussy et al. [13]. Real Time PCR amplifies a 131bp fragment of the ORF75 gene codifying phosphoribosylformylglycinamidine synthase (FGARAT) enzyme. The 7900 HT Fast Real-Time PCR System (Applied Biosystems, Waltham, MA, USA) and SsoAdvanced^TM^ Universal Probes Supermix (BioRad Laboratories, Hercules, CA, USA) were used for this assay.

### 2.5. Absolute OvHV-2 Quantitative PCR

To generate the standard curve, we employed the plasmid pCR 2.1 TA Cloning (Thermofisher-Life Technologies) carrying a fragment of 131 bp of the ORF75 gene codifying FGAM-synthase of OvHV-2, named p-OvHV-2 (10^8^ copie/µL). The cloned fragment was confirmed by Sanger sequencing on both strands by E. Sanger Sequencing 3500 series Genetic Analyzer (Applied Biosystem). The standard curve was created by placing the Cq values of eight serial 10-fold dilutions of p-OvHV-2 (1 × 10^7^ to 1 × 10^1^ copies/µL), performed in triplicate wells, against the log value of the number of copies of p-OvHV-2. The OvHV-2 copy number in each sample was determined using the Cq value to the standard curve and expressed as the number of copies target/µg total DNA extracted. 7900 Software SDS 2.4.1 (Applied Biosystems) generated The Cq value. The efficiency and R2 of the calibration curve were calculated. 

### 2.6. Phylogenetic and Genetic Clustering Analysis

Phylogenetic analyses were performed on the ORF75 gene sequences. Therefore, a dataset (n = 72), which included the sequence isolated during the present study along with all the sequences of the ORF75 gene that are available on GenBank to date, was obtained (see sample codes in figure of the phylogenetic tree below reported for GenBank accession numbers and the geographic origin of the isolates). The sequences included were from Africa (Egypt 2, South Africa 4), Asia (India 9, Iraq 4, Turkey 12), the Americas (Canada 1, USA, 2, Brazil 12), and Europe (Italy 18, Norway 1, Great Britain 6) and were mainly isolated in species belonging to the family of Bovidae, but also in a few individuals belonging to Suidae and Equidae. The sequences’ alignment was carried out using the Clustal Omega package [14] (available at https://www.ebi.ac.uk/Tools/msa/clustalo/ accessed on 29 June 2022). After the alignment, a manual check of the dataset was performed using Unipro UGENE v.35 [15]. The best probabilistic model of sequence evolution was determined by jModeltest 2.1.1 [16], with a maximum likelihood optimized search. A Bayesian phylogenetic tree was obtained using the software MrBayes 3.2.7 [17] with the following model parameters: NST = 6, rates = invgamma, ngammacat = 4. Two independent runs, each consisting of four Metropolis-Coupled MCMC chains (one cold and three heated chains), were run simultaneously for 5,000,000 generations, sampling trees every 1000 generations. The first 25% of the 10,000 sampled trees were discarded as burnin Runs were carried out by means of the CIPRES Phylogenetic Portal [18]. In order to verify the convergence of chains, we checked that the Average Standard Deviation of Split Frequencies (ASDSF), approached 0 [17], and the Potential Scale Reduction Factor (PSRF) was around 1 [19] following Scarpa et al. [20,21], and Casu et al. [22]. The phylogenetic tree was annotated and visualized using FigTree 1.4.1 (available at http://tree.bio.ed.ac.uk/software/figtree/ last access on 3 July 2022). A Principal coordinate analysis (PCoA) was also performed on a K2P [23] pairwise genetic distance matrix using GenAlEX 6.5 [24], to find genetic clusters and/or genetic sub-structuring within clades.

## 3. Results

### 3.1. Clinical and Pathological Findings

The calf showed poor body condition, and signs of neurological disorders including mandibular trismus, tonic-clonic spasm of the muscles and disorientation. Hyperthermia, sialorrhea, hyperaemia with erosive lesions in the buccal mucosae, and bilateral keratoconjunctivitis (Figure 1A) were also observed.

Microscopically, multifocal and generalized mononuclear cell infiltration associated with perivasculitis and necrotizing vasculitis were observed in several sampled organs, including the gastro-intestinal tracts (Figure 1B), kidney, liver, and brain. Moreover, thrombi in the wall of the vessels of the choroid plexus, gliosis and/or neuronal changes were detected in the brain. 

Immunohistochemically, the mononuclear infiltrate cells observed in the perivascular inflammatory foci of the affected organs were mostly characterized by CD3+ T lymphocytes and CD163+ macrophages, whereas CD79+ and PAX5 B cells were few and CD68+ macrophages were not determined. (Figure 2).

### 3.2. Bacteriological Analysis

*Pasteurella* spp. was isolated in the brain and *Streptococcus* spp. was identified by a cultural examination of an ocular swab.

### 3.3. Virological Analysis

The tissues investigated for antigenic presence of BVD, BHV1, BHV4, BRSV, and PI3 resulted negative. We detected the OvHV-2 genome in all of the calf’s organs, including the brain, thymus, lungs, small intestine, kidney, and liver, using Real Time PCR, which was validated by Sanger nucleotide sequencing of amplicons. Results of the target genome copies number/1 µg of total extracted DNA obtained by qPCR in tested tissues organ give a viral load of 2.9 × 10^6^ in lung and 1.8 * 10^6^ in thalamus, while ranging from 1.4 * 10^5^ to 5 * 10^4^ in liver, kidney, and intestine. The Correlation Coefficient of the calibration curve (R2) was 0.99 and the PCR-efficiency of 98%. With regards to the epidemiological investigation in Sardinia, we detected OvHV-2 DNA in all the flocks examined. We observed that 286 out of 1160 samples tested positive for OvHV-2 (24.6%), with a mean farm-level prevalence of 25%. More specifically, the prevalences were 20% in the province of Sassari, 21% in Cagliari, 23% in Oristano, and 33% in Nuoro.

### 3.4. Phylogenetic Analysis

The phylogenetic tree (Figure 3) evidenced a large, well-supported polytomic cluster that included 70 out of 72 of the sequences of the dataset (the Sardinian sequence isolated in the present study was included in this cluster). A mid-point rooting approach was applied to draw the tree and two sequences from North America set as outliers in an external position outside this large polytomic cluster. These latter strains were isolated from bighorn sheep (*Ovis canadensis*) in USA (MN068217) and Canada (KX060582), respectively.

Within the main polytomic cluster, a further well-supported internal sub-cluster was present, with many sequences (70.27%) from North Africa and Asia (from Egypt, Iraq, India, and Turkey, respectively), and only a few (29.73%) from Europe, South Africa, and South America (from Great Britain, South Africa, and Brazil, respectively). Apart from those included in the internal sub-cluster, the other sequences included in this large polytomic cluster, which also included the Sardinian strain isolated in the present study, were almost all from Italy, with only a few from the Americas, Asia, Africa, and Europe (from Brazil, Canada, USA, Turkey, South Africa, Great Britain, Norway, and Turkey, respectively). 

The PCoA (Figure 4) explained the 74.72% of variation (x-axis: 59.11% and y-axis: 15.61%) and was consistent with the phylogenetic tree analysis evidencing two main groups of sequences (groups 1 and 2) separated along the x-axis. In particular, the sequences included in the phylogenetic tree polytomic cluster (with the exception of those within the internal sub-cluster), were grouped within the group 1 of the PCoA whose 54% of sequences were isolated in Italy. Group 2 included all of the sequences included within internal sub-cluster of the phylogenetic tree. Three sequences from India and one from Turkey set outside the group 2 showing a genetic affinity with it on the x-axis. The only outlier was a Brazilian sequence that showed a comparable genetic divergence with both groups 1 and 2.

## 4. Discussion

In our report, histological and virological analysis, as well as phylogenetic inferences, are consistent among each other, thus confirming the presence of OvHV-2 in a calf with the distinctive clinical signs of the disease. To perform a proper histological diagnosis of MFC, representative and specific tissues must be collected during a calf necropsy. In this regard, the sampling of mesenteric lymph nodes, small intestine, lung, eye, carotid rete mirabile, and urinary bladder are typically suggested [25]. Moreover, in outbreaks of MCF, the kidney is one of the organs of choice to be collected for detection of OvHV-2 [25]. Histologically the distinctive signs of the OvHV-2 pathology are infiltrating lymphocytes, which are mostly related to blood vessels and are associated with disseminated necrotizing vasculitis. We revealed these patterns of lesions in multiple tissues, together with lymphocytic portal hepatitis, non-suppurative meningoencephalitis, and lymphocytic interstitial nephritis, which are commonly reported in the literature [25,26,27]. Unfortunately, due to the prolonged fixation, our samples were not suitable to detect OvHV-2 in situ by DNA hybridization in order to additionally confirm its role as aetiologic agent of the lesions. As previously described [2,28], our CD immunophenotyping confirmed that lymphoid cell infiltrations consisted mainly of T cells. Interestingly, we have detected a strong expression of CD163+ macrophages and a total absence of CD68+ macrophages. During inflammation, macrophages switch from a proinflammatory to an anti-inflammatory phenotype promoting the resolution of inflammation and the re-establishment of homeostasis [29], depending on several conditions, including the tissues and the etiological agent involved.

It is broadly accepted that CD163+ macrophages, are considered anti-inflammatory phenotype. Regarding our case, we speculate that OvHV-2 could be capable of inducing anti-inflammatory phenotype in the macrophages of the host for helping the disease progression. Interestingly, a study on lung pathology in man during acute respiratory distress syndrome caused by SARS-CoV-2 revealed an accumulation of CD163+ macrophages which correlated positively with the severity of the disease by intensifying fibro-proliferative response [30].

The evaluation of the number of genomic targets in the calf investigated organs allowed us to highlight that a considerable number of copies of viral genomes are present in the host brain, specifically in the thalamus, slightly less than in the lung, but comparable to those found in the liver, kidney, and intestine. As a result, the high number of target viral genome copies retrieved in our calf’s brain suggests that this organ could also be considered as an elective organ of choice for MCF diagnosis. Our bacteriological results showed that in the case of MCF infection, opportunistic infections from *Pasteurella* spp. and *Streptococcus* spp. can occur, although they seem to be not responsible for the observed symptoms. Although it was not possible to extend the investigation to all animals of the farm where the analyzed calf lived to trace the source of infection, in this study, we found 27 positives out of 98 samples examined (27.5%) in the same geographical area. Phylogenetic and genetic clustering analyses evidenced that the sequence of the gene ORF75 for OvHV-2 isolated in Sardinia is genetically similar to all of the other Italian strains present on GenBank. This finding suggests the possible occurrence of a genetic cluster that is widely distributed across the whole Italian territory. This condition is quite common in DNA viruses as OvHV-2 due to their low mutational rate [31]; on the contrary, RNA viruses generally show higher levels of mutational rate as a consequence of the lack of a quality control mechanism (i.e., proofreading activity of the polymerase) during replication see, e.g., [32].

## 5. Conclusions

In conclusion, this work provides a comprehensive clinical-pathological and virological characterization of a case of MCF in a calf, as well as one preliminary estimate of OvHV-2 prevalence in sheep across the whole Mediterranean island of Sardinia. Given the epidemiological findings on the high circulation of the virus in the territory, which are in contrast to the sporadic clinical detection of the disease in bovine, we believe that further investigation is needed to help understand the risk factors that governing this aspect. Indeed, it is likely that the disease is underestimated and should be included in the differential diagnosis of bovine diseases.

## Figures and Tables

**Figure 1 vetsci-09-00442-f001:**
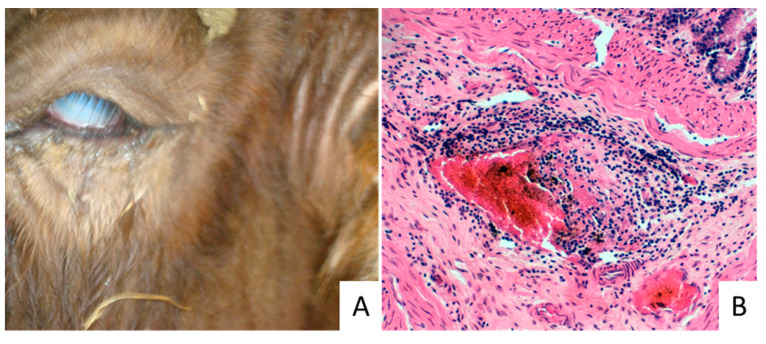
Calf affected by MCF showing keratoconjunctivitis with corneal edema (**A**). Microscopic aspects of intestine in a MCF affected calf. Representative microphotograph of the necrosis affecting a vessel, 20× objective (**B**).

**Figure 2 vetsci-09-00442-f002:**
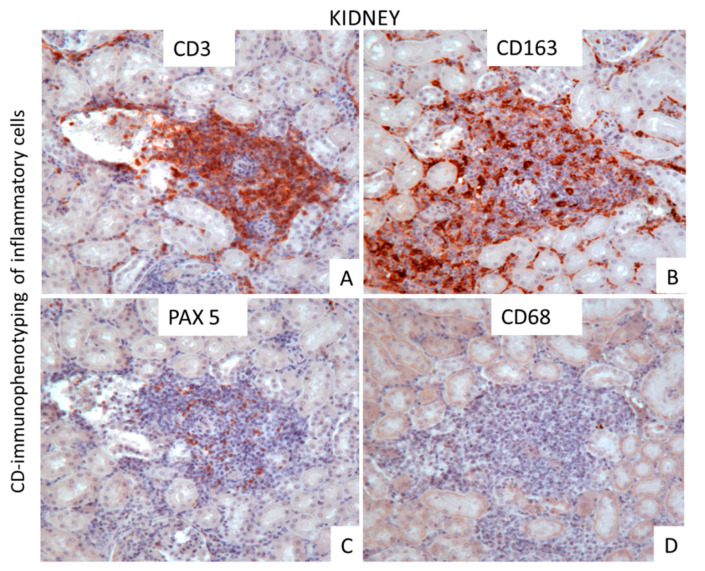
Immunohistochemistry. Kidney (**A**–**D**). CD immune-phenotyping in non-suppurative interstitial nephritis, revealing intense CD3+ T positive (**A**) cells and CD163+ macrophages (**B**), few PAX 5+ B cells (**C**), and no CD68+ macrophages (**D**). 3-30-diaminobenzidine (DAB) chromogen was used to visualize immune reactions. Mayer’s hematoxylin counterstain, 20× objective.

**Figure 3 vetsci-09-00442-f003:**
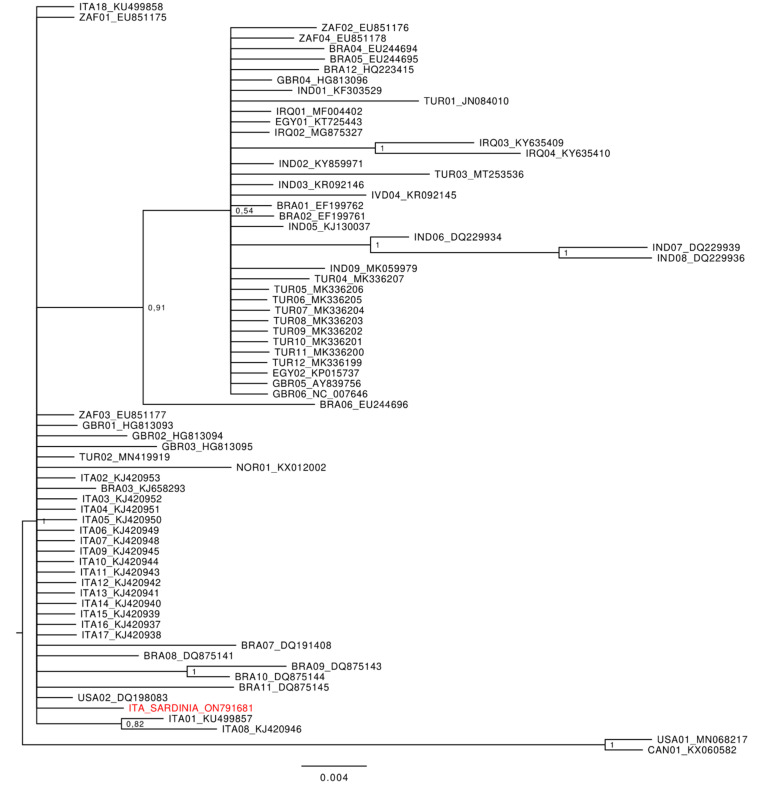
Bayesian phylogenetic tree based on 131bp-long sequences of the ORF75 gene isolated for OvHV-2 in cases of malignant catarrhal fever worldwide. Values of node supports are expressed as Posterior Probabilities. The ORF75 gene sequence isolated in Sardinia is indicated in red font. The codes of sequences within the tree report the conventional code of the country where samples were collected (the first three letters) and the corresponding GenBank accession number.

**Figure 4 vetsci-09-00442-f004:**
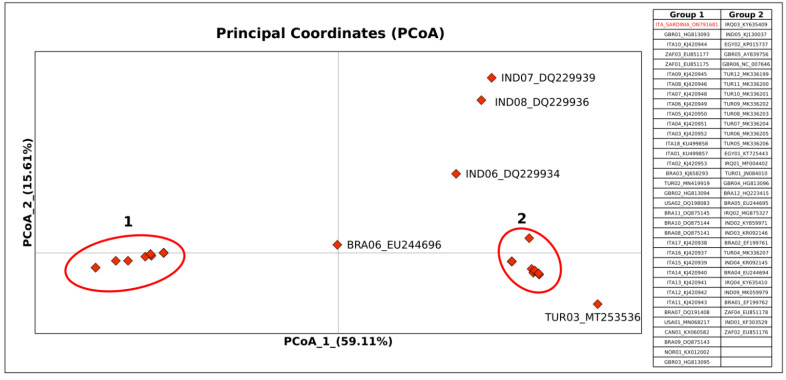
Principal components analysis performed on the same dataset of OvHV2 ORF75 gene fragment (131 bp) used for the phylogenetic tree analysis. Bi-dimensional plots show the genetic differentiation among strains due to the base differences per site found in the dataset. The ORF75 gene sequence isolated in Sardinia is indicated in red font.

## Data Availability

The sequences of OvHV-2 genes obtained during the present study are openly available in the GenBank nucleotide sequence database. Accession numbers: ON791681.

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
