# Peer review of "Malignant Catarrhal Fever in Sardinia (Italy): A Case Report"

_vetsci, 2022, doi:10.3390/vetsci9080442_

Round 1

Reviewer 1 Report

The manuscript "Malignant catarrhal fever in Sardinia (Italy): a case report" by Coradduzza E.. et al., Ref: Manuscript Number: vetsci-1850062, describes a clinical case of malignant catarrhal fever (MCF) that occurred in a calf, which shared the pasture with sheep on a farm located in Sardinia, an Italian island. In order to identify and characterize the virus (ovine herpesvirus type-2, OHV-2), authors have used a multidisciplinary approach, including conventional clinical examination, molecular, histological, and virological methods.

The materials and methods are straightforward to read and follow.

The results are fascinating and reveal that the ORF75 gene sequence of the Sardinian OHV-2 is phylogenetic related to other Italian strains. Including figures and diagrams in completing and clarifying the information, inserting the materials and methods, and the results are highly appreciated.

Overall, the technical and scientific sound of the manuscript seems very good.

The structure and the evaluation of the results underline the importance and relevance of data included in this manuscript and fashioned exquisitely for the reader.

After careful evaluation, I have no comments or criticism to report, and the manuscript is worthy of publication without substantial changes in the Veterinary Science journal.

Author Response

Dear Reviewer 1,

Thank you very much for your revision and advices. Please find below our responses to your suggestions. We used the RED font to evidence the changings that we made in the manuscript according to your comments.

REV1

What was the reason for testing only OvHV-2 by nucleic acid-based method while the rest of the viral agents were screened by IFA?

AUT.

During the study we did not set up PCR protocols for all the diseases we investigated, as the history, the clinical signs and histopathological finding strongly suggested an infection due to OvHV-2 and prompted us to focus on this virus.

REV1

How do the authors evaluate the appropriateness of the 131 pb real-time PCR fragment for the phylogenetic analysis? In GenBank longer sequences are deposited, so an adjusted sequencing approach or targeting another gene might have provided a different resolution and outcome from the obtained polytomic tree? Anyhow, the size of the analyzed sequence portion should be indicated in the legends of the respective figure.

AUT.

We understand the concern of the Reviewer, however the only aim of the phylogenetic analysis in our study was to perform a molecular taxonomic identification of viral strain, which was isolated in Sardinia. Inferences on the relationship among strains from all over the world were out of the goals of our report.

For this reason, we used the short (but highly polymorphic) sequence of the ORF75 fragment obtained for the virus identification at the Real Time to perform also the phylogenetic tree analysis and confirm that the ORF75 gene isolated in Sardinia exactly corresponds to the OvHV-2 which is associated to cases of malignant catarrhal fever worldwide.

In accordance with the request of the Reviewer, we indicated in the legend of figures 4 and 5 the size of the DNA fragment analysed.

REV1

Some language usage/typo issues, e.g. acid nucleic purification (line 91); Border Viral Disease Virus (BVDV).

AUT.

We revised completely the manuscript to assess this issue.

Reviewer 2 Report

This well-written case report presents an intriguing finding based on sound methodology. Some questions regarding the investigations:

What was the reason for testing only OvHV-2 by nucleic acid-based method while the rest of the viral agents were screened by IFA?

How do the authors evaluate the appropriateness of the 131 pb real-time PCR fragment for the phylogenetic analysis? In GenBank longer sequences are deposited, so an adjusted sequencing approach or targeting another gene might have provided a different resolution and outcome from the obtained polytomic tree? Anyhow, the size of the analyzed sequence portion should be indicated in the legends of the respective figure.

Some language usage/typo issues, e.g. acid nucleic purification (line 91); Border Viral Disease Virus (BVDV).

Author Response

Dear Reviewer 2,

Thank you very much for your revision and advices. Please find below our responses to your suggestions. We used the RED font to evidence the changings that we made in the manuscript according to your comments.

REV2

What was the reason for testing only OvHV-2 by nucleic acid-based method while the rest of the viral agents were screened by IFA?

AUT.

During the study we did not set up PCR protocols for all the diseases we investigated, as the history, the clinical signs and histopathological finding strongly suggested an infection due to OvHV-2 and prompted us to focus on this virus.

REV2

How do the authors evaluate the appropriateness of the 131 pb real-time PCR fragment for the phylogenetic analysis? In GenBank longer sequences are deposited, so an adjusted sequencing approach or targeting another gene might have provided a different resolution and outcome from the obtained polytomic tree? Anyhow, the size of the analyzed sequence portion should be indicated in the legends of the respective figure.

AUT.

We understand the concern of the Reviewer, however the only aim of the phylogenetic analysis in our study was to perform a molecular taxonomic identification of viral strain, which was isolated in Sardinia. Inferences on the relationship among strains from all over the world were out of the goals of our report.

For this reason, we used the short (but highly polymorphic) sequence of the ORF75 fragment obtained for the virus identification at the Real Time to perform also the phylogenetic tree analysis and confirm that the ORF75 gene isolated in Sardinia exactly corresponds to the OvHV-2 which is associated to cases of malignant catarrhal fever worldwide.

In accordance with the request of the Reviewer, we indicated in the legend of figures 4 and 5 the size of the DNA fragment analysed.

REV2

Some language usage/typo issues, e.g. acid nucleic purification (line 91); Border Viral Disease Virus (BVDV).

AUT.

We revised completely the manuscript to assess this issue.

Reviewer 3 Report

Here you describe a case of withespread lymphocytic vasculitis associated to the presence of OvHV-2 DNA in different tissues of a calf from Sardinia. However, you have not properly demonstrated that this lessions are produced by the virus. 

Although you have performed a well designed IHQ study to characterize the inflammatory response of the vasculitis, I think this add nothing to the study as such inflammatory reaction has been previously described in this disease. 

I would recomend that IHQ have to centers in the diagnosis of the etiology, i.e. you should perform IHQ against herpesvirus to clearly demonstrate that the virus proteins are present in the lessions, only PCR is not enough to argue a viral induced disease. 

However I think this article provide key epidemiological information for veterinarians who work in Italy and mediterranean coast.  The incidence that you reported is high and merits further analysis. But this information can be published in a much brief document, like a short communication. 

Also other minors corrections are proposed: 

-Lines 146-148. Figure 1 legends' are incorrect.

-Line 160. The legend of the figur2 2 D says that the tissue is kidney when it is liver.

-Figure 2. You are describing vasculitis. However, the pictures that you show are in general of low magnification and I can not clearly appreciate if the inflammation is affecting the vessel wall or, by contrast, this is more a perivascular infiltrate with secondary affection of the vessel. Please, provide some image in which the vasculitis can be clearly appreciated (i.e. inflammatory infiltrate in the vessel wall, necrosis or degeneration of the vessel, etc.).

-Figure 2B. It seems to me that the cut is too thick? Is that cut at 4μm?

-Figure 2 E. I think the constrast color is very prominent in this image, but maybe just my perception.

-What the point of the IHQ analysis? In the literature is well described the inflammatory response of the bovine to OvHV-2. I can no see the point to include this IHQ panel for this case report. That your IHQ results provide new information about the host response to the virus? It is a new presentation not previously described? In the case of negative response, I would suggest to remove the IHQ against this antibodies. It is a good work, but adds nothing new to your report. 

 -For my point of view, the figure 4 is not necessary.

Author Response

Dear Reviewer 3:

Thank you very much for your revision and advices. Please find below reported our responses to your suggestions. We addressed all your requests and strongly revised the manuscript accordingly. We used the BLUE font to evidence the changings that we made in the manuscript according to your suggestions.

REV3

Here you describe a case of withespread lymphocytic vasculitis associated to the presence of OvHV-2 DNA in different tissues of a calf from Sardinia. However, you have not properly demonstrated that this lessions are produced by the virus.  Although you have performed a well designed IHQ study to characterize the inflammatory response of the vasculitis, I think this add nothing to the study as such inflammatory reaction has been previously described in this disease. I would recomend that IHQ have to centers in the diagnosis of the etiology, i.e. you should perform IHQ against herpesvirus to clearly demonstrate that the virus proteins are present in the lessions, only PCR is not enough to argue a viral induced disease.

AUT.

We agree with the Reviewer’s suggestion to perform IHQ against herpesvirus, indeed during our study we try to determined it using RNAscope probe directed against OvHV-2 (Pasavento et  al., Vet Pathol. 2019 Jan;56(1):78-86.) without successful results. Our negative results were exclusively caused by the formaline over-fixed samples which make less efficient the retrieval treatment. Clinical signs and gross pathology are extremely variable and not pathognomonic, while histological examination is a well-known useful diagnostic tool when performed on different tissues, particularly when coupled with the viral DNA. In addition, the presence of number of other possible ethiological agents has been excluded in the same organ by IFA.  

REV3

What the point of the IHQ analysis? In the literature is well described the inflammatory response of the bovine to OvHV-2. I can no see the point to include this IHQ panel for this case report. That your IHQ results provide new information about the host response to the virus? It is a new presentation not previously described? In the case of negative response, I would suggest to remove the IHQ against this antibodies. It is a good work, but adds nothing new to your report.

AUT.

We agree with the Reviewer’s sentence “In the literature is well described the inflammatory response of the bovine to OvHV-2”, nevertheless, studies are routinely focused on the characterization of the T cells population (Anderson et al., 2007). Our IHQ was finalized to better describe the other two inflammatory cells we observed (B cells and macrophage). We tested two antibodies for B cells (CD79 and Pax 5) and CD163 and CD68 antibodies for macrophages determination. The results show that different markers of activation/differentiation are expressed in monocytes/macrophages population. In this respect we add a new sentence in the manuscript LINE 232-235 “Interestingly, we have detected a strong expression of CD163+ macrophages and a total absence of CD68+ macrophages. These results indicate that different markers are expressed in monocytes/macrophages population, although their role in the pathogenesis of MCF remain to be understood”.  For all these reasons, we would prefer to maintain also our IHQ in the manuscript.

REV3

However I think this article provide key epidemiological information for veterinarians who work in Italy and mediterranean coast.  The incidence that you reported is high and merits further analysis. But this information can be published in a much brief document, like a short communication.

AUT.

We understand the concern of the Reviewer, but, based on the 1160 samples used for prevalence analysis, in the next future we are going to write a manuscript in which the polymorphisms of 4 genes encoding 3 glycoproteins and 1 tegument protein will be characterized. Phylogenetic analyses will allow the description of the strains currently distributed in Sardinia, further expanding our knowledge on this topic.

REV3

Lines 146-148. Figure 1 legends are incorrect.

AUT.

We have corrected the legend of the Figure 1.

REV3

Line 160. The legend of the figure 2 D says that the tissue is kidney when it is liver.

AUT.

We modified the legend of figure 2.

REV3

Figure 2. You are describing vasculitis. However, the pictures that you show are in general of low magnification and I can not clearly appreciate if the inflammation is affecting the vessel wall or, by contrast, this is more a perivascular infiltrate with secondary affection of the vessel. Please, provide some image in which the vasculitis can be clearly appreciated (i.e. inflammatory infiltrate in the vessel wall, necrosis or degeneration of the vessel, etc.).

AUT.

We modified the Figure n. 2.

REV3

Figure 2B. It seems to me that the cut is too thick? Is that cut at 4μm?

AUT.

The slide is cut at 4μm, however, we agree with the Reviewer’s concern and we deleted the figure. See THE NEW PANNEL PROPOSED FOR FIGURE 2.

REV3

Figure 2 E. I think the constrast color is very prominent in this image, but maybe just my perception.

AUT.

We agree with the Review’s perception and we improved the quality of the image (SEE THE NEW PANNEL PROPOSED FOR FIGURE 2).

REV3

For my point of view, the figure 4 is not necessary.

AUT.

We agree with the Reviewer and removed this figure.

Round 2

Reviewer 3 Report

Dear authors, 

After the first revision, I have more suggestions that can be useful to improve the quality of your scientific work. 

As I said in the previous revision, I believe that the case report you are describing of infection by OvHV-2 in a calf is not new et all, and similar cases can be found widespread in the literature. However, you performed an immunohistochemical panel and found that inflammatory reaction centred in the lesions (presumably attributed to the infection) is composed mainly of T-cells (as already known) and CD163+ macrophages (not reported in this condition). I think this finding can be considered interesting and agree with the authors that merit further investigation. 

I also think, like the authors, that the described lesions are most probably attributed to the infection with OvHV-2, but unfortunately, science is not about believing but about demonstrating facts by validated methods. You have undoubtedly demonstrated the presence of the viral DNA in your samples, but you have not demonstrated that the virus is actually in the lesions. As I asked in the first revision, the IHQ study must include markers against herpesvirus. Alternatively, ISH or immunofluorescence can be performed. If you can not demonstrate the viral agent within the lesions, the case report can be removed and the article can be centred on the molecular results. 

How can we be sure that the inflammatory pattern that you describe with your IHQ is not attributed to other non-tested pathogens like parasites or other viral conditions? And if we can not be sure of that, what's the point to perform such a good panel to characterize the immune response? In my opinion, if you want to publish the complete case report including the IHQ results you need to confirm the presence of the virus in the lesions, otherwise your interpretations of your IHQ can not be valid. 

I have a few more suggestions: 

- Figure 1 and figure 2 can be condensed into only one panel. Figure 1 A can be removed. In figure 2 you reported lesions that are not new and most of them not really specific of this condition. I would recommend choosing only one  (maybe the vasculitis).

- The figure 3 can stay, but its relevance in the absence of IHQ against HV is questionable. 

- In general, I found the histological images of low quality, but maybe is my version of the paper. Recheck that you meet the figures criteria recommended by the journal for all the figures.

- The discussion and conclusion can be improved (see the revised document). 

Author Response

Dear Reviewer 3:

Thank you very much for your revision and advices. Please find below reported our responses to your suggestions. We addressed all your requests and revised the manuscript accordingly. We used the BLUE font to evidence the changings that we made in the manuscript according to your suggestions.

REV3

After the first revision, I have more suggestions that can be useful to improve the quality of your scientific work.As I said in the previous revision, I believe that the case report you are describing of infection by OvHV-2 in a calf is not new et all, and similar cases can be found widespread in the literature. However, you performed an immunohistochemical panel and found that inflammatory reaction centred in the lesions (presumably attributed to the infection) is composed mainly of T-cells (as already known) and CD163+ macrophages (not reported in this condition). I think this finding can be considered interesting and agree with the authors that merit further investigation. I also think, like the authors, that the described lesions are most probably attributed to the infection with OvHV-2, but unfortunately, science is not about believing but about demonstrating facts by validated methods. You have undoubtedly demonstrated the presence of the viral DNA in your samples, but you have not demonstrated that the virus is actually in the lesions. As I asked in the first revision, the IHQ study must include markers against herpesvirus. Alternatively, ISH or immunofluorescence can be performed. If you can not demonstrate the viral agent within the lesions, the case report can be removed and the article can be centred on the molecular results.How can we be sure that the inflammatory pattern that you describe with your IHQ is not attributed to other non-tested pathogens like parasites or other viral conditions? And if we can not be sure of that, what's the point to perform such a good panel to characterize the immune response? In my opinion, if you want to publish the complete case report including the IHQ results you need to confirm the presence of the virus in the lesions, otherwise your interpretations of your IHQ can not be valid. 

AUT.

We agree with the referee and we insert, in the discussion paragraph of the manuscript, the following sentence ( lines: 229 – 231 ) in order to inform the reader on this weakness of the study: “ Unfortunately,  because of the prolonged fixation, our samples were not suitable to detect OvHV-2 in situ by DNA hybridization in order to additionally confirm its role as aetiologic agent of the lesions.”

REV3

Figure 1 and figure 2 can be condensed into only one panel. Figure 1 A can be removed. In figure 2 you reported lesions that are not new and most of them not really specific of this condition. I would recommend choosing only one  (maybe the vasculitis).

AUT.

We change the manuscript according to the suggestion.

REV3

The figure 3 can stay, but its relevance in the absence of IHQ against HV is questionable. 

AUT.

We confirm the figure 3 (now 2) but we improve the discussion about the results (see the manuscript).

REV3

In general, I found the histological images of low quality, but maybe is my version of the paper. Recheck that you meet the figures criteria recommended by the journal for all the figures.

AUT.

We follow the criteria recommended by the journal

REV3

The discussion and conclusion can be improved (see the revised document). 

AUT

We have revisited the discussion and conclusion, see the manuscript.